# Customizable Three-Dimensional Printed Earring Tap for Treating Affections Caused by Aesthetic Perforations

**DOI:** 10.3390/pharmaceutics16010077

**Published:** 2024-01-05

**Authors:** Ludmila A. G. Pinho, Ana Luiza Lima, Yong Chen, Livia L. Sa-Barreto, Ricardo N. Marreto, Guilherme M. Gelfuso, Tais Gratieri, Marcilio Cunha-Filho

**Affiliations:** 1Laboratory of Food, Drugs, and Cosmetics (LTMAC), University of Brasilia, Brasilia 70910-900, DF, Brazil; ludmila.alvim@gmail.com (L.A.G.P.); 220007667@aluno.unb.br (A.L.L.); liviabarreto@unb.br (L.L.S.-B.); gmgelfuso@unb.br (G.M.G.); tgratieri@gmail.com (T.G.); 2Laboratory for Drug Delivery & Translational Medicine, School of Pharmacy, Nantong University, 19 Qixiu Road, Nantong 226001, China; scuchen2003@ntu.edu.cn; 3Laboratory of Nanosystems and Drug Delivery Devices (NanoSYS), School of Pharmacy, Federal University of Goias, Goiania 74605-170, GO, Brazil; ricardomarreto@ufg.br

**Keywords:** aesthetic perforation, pharmaceutical jewelry, wearable devices, FDM printing, drug topical delivery

## Abstract

This work aimed to develop a three-dimensional (3D) wearable drug-loaded earring tap to treat affections caused by aesthetic perforations. The initial phase involved a combination of polymers to prepare filaments for fused deposition modeling (FDM) 3D printing using a centroid mixture design. Optimized filament compositions were used in the second phase to produce 3D printed earring taps containing the anti-inflammatory naringenin. Next, samples were assessed via physicochemical assays followed by in vitro skin permeation studies with porcine ear skin. Two filament compositions were selected for the study’s second phase: one to accelerate drug release and another with slow drug dissolution. Both filaments demonstrated chemical compatibility and amorphous behavior. The use of the polymer blend to enhance printability has been confirmed by rheological analysis. The 3D devices facilitated naringenin skin penetration, improving drug recovery from the skin’s most superficial layer (3D device A) or inner layers (3D device B). Furthermore, the devices significantly decreased transdermal drug delivery compared to the control containing the free drug. Thus, the resulting systems are promising for producing 3D printed earring taps with topical drug delivery and reinforcing the feasibility of patient-centered drug administration through wearable devices.

## 1. Introduction

Three-dimensional printing (3DP) is a set of technologies that potentially brings relevant changes to the pharmaceutical field [1]. Among the main categories of 3DP techniques, including vat photopolymerization, directed energy deposition, powder bed fusion, binder jetting, material jetting, and sheet lamination, material extrusion seems to be one of the most promising for pharmaceutical applications [2,3].

Notably, fused deposition modeling (FDM) is a thermal material extrusion technology that has been highly popularized due to its low cost, versatility, and ability to produce products with high mechanical strength [4]. Indeed, numerous drug delivery devices have been made by FDM 3DP, including tablets with immediate or controlled drug release, vaginal rings, orthodontic retainers, drug implants, catheters, nasal supports, and even devices that can treat prosthetic infections, a therapeutic intervention fraught with numerous challenges within contemporary medical and pharmaceutical domains [5,6,7,8].

Considering the digital revolution in the last 25 years, the one-size-fits-all approach to medicine is outdated as it fails to meet the market’s needs. An exciting aspect of 3DP is the capability to provide personalized therapeutic medicines that meet each patient’s physiological and lifestyle needs [9]. Furthermore, such technology allows the production of patient-centered drug devices such as tablets with Braille designs for the visually impaired, multi-drug-loaded hearing aids with anti-biofilm properties, and microdevices with stimuli-responsive release mechanisms [10], which not only promote dose personalization but also increase patient medication adherence [11].

Infectious and inflammatory processes arising from the use of adornments are a wide-ranging problem that, to this day, lacks satisfactory treatment. Indeed, it has been estimated that 35% of individuals who wear earrings have one or more complications. The treatment usually requires removing the earring or piercing, which leads to healing and closing the perforation made to fit the adornment [12,13]. Recognizing the limitations of conventional approaches, recent advancements in medical technology have introduced the potential of 3D printed devices to address complications arising from prosthetic infections [7,8]. In fact, navigating the landscape of pharmaceutical interventions for addressing prosthetic infections in the contemporary era brings forth an array of challenges and intricacies, necessitating innovative resolutions enabled by the advancements in 3D printing technology.

Recent research has explored the idea of pharmaceutical jewelry and proposed a transdermal drug delivery system containing a hormone for chronic treatment [14]. Yet, there are no records of personalized therapeutic 3D systems aimed at a topical treatment, which could benefit many jewelry users with a more convenient recovery without the drawback of losing the perforation made to fit the jewelry.

Here, we develop an innovative FDM 3DP wearable drug-loaded earring tap to treat or prevent the affections caused by aesthetic perforations. This proof-of-concept study exploits a device that might be used for either earring affections or any other disease that would need topical treatments since 3DP allows multiple printing shapes. Due to the difficulty of fabricating filaments with suitable mechanical characteristics, melt-blending can be an affordable and convenient solution for the FDM process [15]. Accordingly, a mixture design was used to find a polymer blend with adequate matrix solubility and printability using an anti-inflammatory model drug, naringenin (NAR) [16]. Finally, mechanical and physicochemical tests were carried out to characterize such systems, followed by in vitro skin permeation studies.

## 2. Materials and Methods

### 2.1. Material

NAR ((2S)-5,7-dihydroxy-2-(4-hydroxyphenyl)-2,3-dihydro-4H-chromen-4-one, purity ≥ 98%, lot MKCD1056) was obtained from Sigma-Aldrich (St. Louis, MO, USA). The polymer Parteck^®^ MXP (PVA, polyvinyl alcohol, lot F1952064) was donated by Merck (Darmstadt, Germany). Klucel^TM^ EF (HPC, hydroxypropylcellulose, lot 40915) and Soluplus^®^ (SOL, polyvinyl-caprolactam-polyvinyl-acetate-polyethyleneglycol, lot 844143368EO) were donated by Ashland Specialty Ingredients (Covington, LA, USA) and BASF (Ludwigshafen, Germany), respectively. Glycerin (GLY, lot 58591) and Tween 80 (polysorbate, lot 105896) were obtained from Dinâmica^®^ (Sao Paulo, Brazil). Skin from porcine ears was obtained from a local slaughterhouse (Via Carnes Indústria e Comércio, Brasilia, Brazil). The whole skin was removed from the outer region of the ear, separated from its underlying layer, and used full thickness.

### 2.2. Mixture Design

The HME filaments were prepared according to a simplex centroid mixture design with three components without constraints to determine the ideal combination of selected biocompatible polymers (Table 1). The responses obtained for printability (fracture force), drug release (dissolution efficiency), and moisture absorption were analyzed using the software Design Expert 11.0 (Stat-Ease, Minneapolis, MN, USA). The possible mathematic models were examined using a one-way analysis of variance (ANOVA). The best-fitting model was selected for each response based on *p*-values, and the predictive equations containing only significant terms were built from stepwise multiple regression analysis [17]. Then, two filaments with different characteristics were produced using optimized models. Specifically, one model was produced with the lowest moisture absorption (importance +), higher fracture force (importance ++), and higher dissolution efficiency (importance +++), and another model was produced with the same parameters but a dissolution efficiency of 40% (importance +++). Once the data were inserted, the software chose the best polymer combination to meet the expected responses. The correlations among the predicted and the experimental values were evaluated.

### 2.3. The Preparation of Filaments by Hot-Melt Extrusion (HME)

Filaments were prepared according to Table 1 using 15% GLY as a plasticizer and 15% NAR. The mixtures were prepared using mortar and pestle and then extruded without recirculation in a co-rotating conical twin-screw extruder HME with a die diameter of 1.8 mm (HAAKE MiniCTW, ThermoScientific, Waltham, MA, USA) coupled to a filament tractor with an air-cooling system and an automated diameter measurement model FTR1 (Filmaq3D, Curitiba, Brazil). The extrusion conditions were chosen to guarantee smooth flow and uniform filament diameter. All filaments were stored in a desiccator until characterization.

### 2.4. Tensile Test

A tensile test was performed to identify the filaments’ fracture force (*n* = 5) and predict printability via the mechanical resistance of filaments [18]. All analyses were executed in a universal testing machine (Shimadzu EZ test, Tokyo, Japan) equipped with a 5 kN load cell using wedge-type grips that move horizontally to tighten the grip on the filament before the analyses and vertically to perform the elongation test. The cell moved at a constant crosshead speed of 10 mm min^−1^ [18]. The filament size was 40 mm, the gap between the cells was 20 mm, and the initial force was 0 N.

### 2.5. Moisture Absorption

To predict the filaments’ moisture absorption, each filament’s water content was assessed using TGA by heating the samples until 105 °C and holding this temperature for 60 min under a nitrogen atmosphere (flow rate of 50 mL min^−1^) [19]. Then, the filaments were stored in a desiccator with an NaCl-saturated solution at 40 °C (relative humidity 75%) for 7 days, and the moisture content was re-evaluated. The difference between the initial and the final moisture was considered the moisture uptake.

### 2.6. Dissolution Studies

The dissolution profiles of NAR and filaments were determined in an Ethik dissolution apparatus model 299 (Nova Ética, Sao Paulo, Brazil) using 900 mL of HCl 0.1 mol L^−1^ as dissolution medium. Temperature was maintained at 37 °C, and apparatus 2 (paddle) operated at 100 rpm. Samples containing 25 mg of the drug were added to the dissolution vessel. Aliquots of 5 mL were withdrawn and immediately replaced by fresh dissolution medium at 5, 10, 15, 20, 30, 45, 60, 120, 180, 240, 300, 360 min, and 24h. After filtration (0.45 µm), samples were taken to a spectrophotometer for NAR content quantification as described in Section 2.15 [16]. Experiments were performed in triplicate for each sample, and dissolution profiles were evaluated using their corresponding dissolution efficiency at 24h (DE24). Statistics were evaluated using GraphPad Prism 8 software (San Diego, CA, USA) using one-way ANOVA, followed by Tukey post-test. The significance level (*p*) was fixed at 0.05. Data normality was previously demonstrated using the Shapiro–Wilk normality test.

### 2.7. Earring Devices Produced by FDM 3DP

Cylinders of 13 mm in diameter and 1 mm in height were graphically designed and sliced using free versions of Tinkercad^®^ (https://www.tinkercad.com/, Autodesk^®^ Inc., San Rafael, CA, USA) and Slic3r^®^ (version 1.3.0, Rome, Italy) software, respectively. FDM Voolt 3D model Gi3 printer (Sao Paulo, Brazil) endowed with a nozzle diameter of 0.4 mm was used to print the filaments chosen in Section 2.2. The printing temperature was 185 °C for the first layer and 180 °C for other layers. The temperature of the building platform was set at 65 °C, and five devices were printed at a time. The layer height was fixed at 0.2 mm, the infill density was 10% using a rectilinear pattern, and the printing speed was 15 mm s^−1^ for printing moves and 50 mm s^−1^ for travel speed.

### 2.8. Thermogravimetric Analysis (TGA)

TGA was obtained using a DTG-60H (Shimadzu, Kyoto, Japan) in platinum pans at a heating rate of 10 °C min^−1^ from 25 to 500 °C. All analyses were conducted under a nitrogen flow of 50 mL min^−1^. Tests were performed on individual compounds, physical mixtures, filaments, and printed devices (previously milled by a knife mill). Assays were carried out using the TA software (version 2.20, Shimadzu, Kyoto, Japan), and the first derivative TGA curves were plotted using the OriginPro software (version 2022, Originlab Corp., Northampton, MA, USA).

### 2.9. Fourier Transform Infrared Spectroscopy (FTIR)

Individual compounds, physical mixtures, filaments, and printed devices (previously milled by a knife mill) were analyzed by FTIR performed in a Vertex 70 FTIR spectrometer using an ATR imaging accessory (Bruker, Billerica, MA, USA). Spectra were recorded between 4500 and 400 cm^−1^ at an optical resolution of 2 cm^−1^.

### 2.10. X-ray Powder Diffraction (XRPD)

XRPD spectra of individual compounds, physical mixtures, and printed devices were collected using a D8 FOCUS XRPD (Bruker, Billerica, MA, USA). The scan speed was 2° min^−1^, and the step size was 0.02°. The diffraction patterns were obtained at angles between 5 and 60° (θ–2θ).

### 2.11. Rheological Analysis

Rheological studies were conducted with the filaments using a DHR-2 rheometer (TA instruments, New Castle, DE, USA) equipped with a Peltier plate and a 40 mm parallel plate geometry, with a gap height fixed at 8000 μm. Small amplitude oscillatory shear was accomplished to ensure that the tests were conducted in the linear viscoelastic range defined by an amplitude oscillatory sweep at a strain (γ) range from 0.001 to 1% at a constant angular frequency of 1 Hz at the maximum and minimum temperature of analysis, i.e., 185 °C and 25 °C. All samples showed a linear response at γ = 0.01%; hence, this deformation amplitude was chosen for the subsequent analysis.

A temperature ramp determined from 185 to 25 °C at a heating rate of −3 °C min^−1^ with ω = 6.28 rad s^−1^ was used to determine the storage (G′), loss (G″), and complex moduli (G*), and the complex viscosity (η*) was determined by:
η* = G*/ω

### 2.12. Morphological Analysis

The samples’ morphological characteristics were evaluated by optical microscopy using a stereoscope coupled to a video camera (Laborana/SZ–SZT, São Paulo, Brazil) and a scanning electron microscope (SEM, Jeol, JSM-7001F, Tokyo, Japan). For SEM analysis, samples were previously sputter-coated with gold using a vacuum sputter coater (Leica EM SCD 500, Wetzlar, Germany).

### 2.13. Drug Release

The in vitro NAR release profile was determined using modified Franz-type diffusion cells. Donor and receptor chambers were separated by a dialysis tubing cellulose membrane (Sigma-Aldrich, Steinheim, Germany). Printed devices containing 25 mg of NAR were added to the donor chamber. The receptor chamber was filled with 15 mL of ethanol at 60% (Sink condition). The release kinetic was carried out with NAR determination in the receptor solution medium for 24 h using a spectrophotometer for NAR content quantification as described in Section 2.15 in five replicates per sample [16]. The statistical analysis was performed by GraphPad Prism 7 (GraphPad Inc., San Diego, CA, USA). All results showed parametric behavior and were compared using one-way ANOVA, followed by a Tukey post-test. The significance level (*p*) was fixed at 0.05.

### 2.14. In Vitro Skin Permeation

In vitro permeation experiments used Franz-type diffusion cells mounted with porcine ear skin. The receptor compartment was filled with 15 mL of 1% Tween 80 (polysorbate) aqueous solution. The donor compartment was wetted with 0.1 mL of distilled water before placing the printed devices (printed devices area = 1.33 cm^2^). Plastic inert disks containing an equivalent amount of NAR were used as control. Plastic disks were produced by imbibing an acetone drug solution (100 mg mL^−1^) on their surface, followed by 30 min of drying. The diffusion cells were maintained at 37 ± 2 °C under magnetic stirring (300 rpm) for 24h (*n* = 5).

At the end of the experiments, the receptor solution was withdrawn and analyzed for NAR content. Then, the skin was placed on a flat surface with the stratum corneum facing upwards. The skin’s stratum corneum was removed with 15 adhesive tapes. Finally, the remaining skin was perforated with the aid of a scissor. The adhesive tapes and the skin fragments were placed in individual glass containers with 5 mL of methanol and left for 24h under magnetic stirring for drug extraction [20]. The resulting aliquots were filtered with 0.45 μm, and the recovered NAR was quantified using the chromatographic method described in the section below. Statistical analysis was performed for each skin layer using GraphPad Prism 8 (GraphPad Inc., San Diego, CA, USA). Results had parametric behavior on the Shapiro–Wilk test. Next, they were compared using one-way ANOVA, followed by a Tukey post-test.

### 2.15. Drug Determination

NAR was quantified using a reversed-phase chromatographic method with detection at 290 nm using the high-performance liquid chromatograph model LC-20AT (Shimadzu, Kyoto, Japan). The operating conditions of the method were as follows: 10 µL of injection volume; reversed-phase C18 column (LC Column, 300 × 3.9 mm, 10 μm); methanol/phosphoric acid 0.01 mol L^−1^ (65:35, *v*/*v*) as the mobile phase; flow rate of 0.6 mL min^−1^; and temperature of 40 °C. The method was previously validated for NAR recovered from the skin [20].

Otherwise, a UV-1800 spectrophotometer (Shimadzu, Kyoto, Japan) was set at 288 nm for NAR determination in dissolution assay. The linearity showed a correlation coefficient (r) of 0.999 in the range of 2.5 to 15 µg mL^−1^, with the slope different from zero, and the residues were randomly distributed without tendency (y = 0.061x − 0.0038).

## 3. Results and Discussion

NAR is feasible for topical use once its Log P (2.4) and low molecular weight (272.3 Da) favor skin absorption. Besides, its low oral bioavailability (around 5.8%) justifies the choice of an alternative administration route. NAR also has favorable characteristics for thermal processes of pharmaceutical production (initial degradation temperature of 207 °C). Nevertheless, its low solubility in many polymers used in FDM 3DP can be a limiting factor [16,20]. Therefore, the study’s first phase used a mixture design approach to find the optimum polymer combination for NAR topical processing. In fact, the complex relationship between experimental variables in HME makes the use of experimental design a highly recommended strategy for directing the mechanical and physicochemical properties of extrudates, which is of particular importance for filaments that will feed 3D printers [21].

The results showed that the combination of polymers could produce uniform and opaque filaments with a yellowish aspect characteristic of NAR (Table 1). On the other hand, the filaments produced just with SOL (HME 3) had a translucid aspect, denoting an intense interaction between this polymeric matrix and the drug [16]. Still, this polymer produced too brittle filaments that could not be measured in fracture force assays (pre-test failure). In fact, SOL filaments were previously reported to be unprintable even with the aid of plasticizers [22]. The filaments containing PVA in turn (HME 1, HME 4, HME 6, and HME 7) had better performance in the fracture test, expressed by values of fracture within the range of 2.5 and 8.5 N mm^−1^, which were described as printable (Figure 1A) [23], indicating that this polymer might contribute to extrudates’ printability [19]. On the other hand, HPC filaments were mostly elastic, which means they took longer to break but had a lower fracture force than PVA [24]. Nonetheless, using a polymer blend seems to positively impact polymer printability, indicating that combining different materials might have a synergic effect on stress resistance, improving printability and increasing the number of materials that can be used for 3DP [25].

Water is known to act as a plasticizer, leading to a change in the filaments’ viscoelastic properties, which could result in significant stability problems for such systems. Filaments’ water uptake was then assessed to find the lower hygroscopic polymer blend [19]. The initial water content ranged from 1.9 to 6.9% after fabrication. Despite the differences in hygroscopicity between the studied polymers, all had a final water content above 10%, emphasizing the need for a sealed package for storage. The high water uptake decreases transition temperature, leading to 3DP failures, and hampers its use as a shelf-item product for on-demand manufacturing [26]. Mixing PVA with SOL or PVA with HPC tends to decrease water capture (Figure 1B).

In dissolution tests (Figure 1C), SOL filaments were shown to decrease the drug dissolution rate, as already observed in a previous study [16]; however, systems containing only PVA or HPC had higher DE24, showing the ability of such polymers to increase drug dissolution and their potential to be used for immediate release systems. Moreover, the polymer blend had an intermediate drug release, showing that it can modulate the drug dissolution as desired using appropriate polymer blends.

Based on the optimized response considering these three responses analyzed, two filaments were proposed to continue the study. Their composition is described in Table 2.

Filaments were chosen according to their printing performance, low water uptake, and controlled or maximized NAR dissolution. Filament A was produced with PVA, while filament B used a polymeric blend. For evaluating the impact of processing and matrix-drug compatibility, samples were assessed before any treatment (PM), after HME processing (HME), and after printing (3D device). TGA results (Figure 2) showed four decomposition phases, the first corresponding to a water loss of around 3%, followed by GLY decomposition (GLY, as supplied, has single-phase decomposition between 123.7 °C and 268.6 °C with 95.96% of weight loss) and NAR:polymer decomposition. TGA results showed a profile compatible with the sum of single compounds. No sign of incompatibility or loss of stability was found.

No discernible differences in degradation curves were observed between the physical mixture, extruded filament, and 3D-printed devices. This observation suggests that the thermal and mechanical stresses from extrusion, along with the subsequent thermal stress during printing, did not induce degradation or any consequential interaction that might have adversely affected the components of the sample. Therefore, under the processing conditions used, the formulation demonstrated thermal stability without any signs of incompatibility or material loss.

In FTIR analysis (Figure 3), the characteristics bands of NAR hydroxyl at 3261 cm^−1^, C=O at 1586 cm^−1^ and 1496 cm^−1^, C-O-C vibration at 1246 cm^−1^, 830 cm^−1^ vibration from the para-substituted aromatic ring, and the CH_2_ stretch on 2933 cm^−1^ from GLY are highlighted and shown to be present in all samples with some slight changes in intensity [27]. In fact, the high similarity between the spectra during all processing steps indicates the chemical stability of the systems.

XRPD results (Figure 4) showed that the characteristic diffraction peaks of NAR were present, although discreet, before any treatment. However, after printing, the systems seem to have become entirely amorphous. Although XRPD identified no crystals, the printed samples’ SEM images (Figure 5) showed that the devices produced with filament B had small crystals on their surface. This behavior might be related to the technique’s limitation in identifying small traces of crystallinity or nanocrystals in a polymer matrix [28]. In fact, those crystals are smaller than NAR as supplied, which presents a crystal size of around 40 µm, while such crystals had approximately 12–15 µm, forming clusters all over the printed surface object. No crystal was seen inside or outside the printed devices for filament A, which showed a smooth and uniform surface.

The drug release of the printed devices (Figure 6) showed no difference between the samples and control when the area under the curve was compared (*p* < 0.05). The release of NAR could occur via the diffusion of the solvent through the polymeric matrix and swelling or relaxation of the polymer chains. Such a process makes the drug diffusion comparable to that of the solubilization of the drug as supplied and used as a control. Although 3D devices A and B have different solubilization capacities, in the drug release assay, the limited free water and the direct contact of the devices with the membrane delay their disintegration, leading to similar drug release kinetics.

The rheological analysis of the molten material is typically used to evaluate the printability of polymeric filaments in an FDM 3D printer [18]. Specifically, the printability assessment involves monitoring the complex viscosity and storage and loss moduli during the printing temperature. The filaments should have a balance between flexibility and brittleness to feed an FDM 3D printer properly. If they are too flexible, they will be squeezed between the feeding gears and cannot be pushed through the printer nozzle. On the other hand, brittle filaments will be easily broken by feeding gears [9]. Moreover, physical modifications to the filament, such as the crystallization of components, can jeopardize its printability. Consequently, the rheological properties of the extruded formulations were measured and correlated with printability in this study.

Although SOL has been reported to be unprintable due to excessive brittleness even with a low incorporation of plasticizer [24], its use in a polymer blend with HPC and PVA showed adequate flow behavior (Figure 7) and uniformity in printing. In fact, both devices were smoothly printed with consistent weight: 176.0 ± 1.5 mg for filament A, and 169.5 ± 2.4 mg for filament B. The difference in weight between filaments is due to the density of the filaments since the same file was used for both printings. This smooth flow is expressed by the low complex viscosity in both samples in the printing temperature, which some authors suggest to be below 8000 Pa.s to promote proper flow and the correct functioning of the printer, and both filaments achieve this mark around 154 °C for filament A and 129 °C for filament B.

Another important rheological parameter is the crossover point (when G′ = G″), which indicates a transition from a more solid-like to a more fluid-like viscoelastic behavior or vice versa [29]. The crossover temperatures of the samples were above 155 °C. This revealed that below this temperature, the storage modulus is higher than the loss modulus, which indicates a more rigid filament at room temperature and an appropriate polymer recovery after fused deformation and the production of suitable 3DP drug products [18]. Thus, the rheological analysis revealed the filaments’ suitability and highlighted the importance of the simple mechanical screening method in the study’s first phase for achieving printable filaments [22].

In vitro skin permeation studies (Figure 8) showed the 3D devices increased NAR skin accumulation compared to the control, regardless of the formulation composition. NAR physicochemical characteristics make it a good permeant through biological membranes, as can be observed by the drug permeation amounts through the skin into the receptor medium after 24 h from an inert control plastic disc, from which the drug was solely released, and there were not any formulation excipients interacting with the skin or altering its barrier properties (Figure 8C). However, a higher skin accumulation is desired over a higher transdermal permeation for a local cutaneous effect.

Ideally, the formulation should target the drug delivery to the superficial skin layers, where it is expected to exert its action and, therefore, prevent dermal exposure. Strategies for this may include controlling drug release or using formulation excipients that interact with skin components and alter its organization and lipophilicity, changing the drug partitioning behavior. Drug release results showed that despite the inclusion of SOL and HPC in a polymeric blend (3D device B), drug release was not controlled compared to either 3D device A or the inert plastic control (Figure 6). Yet, both printed devices significantly altered drug accumulation in the skin.

Specifically, 3D device A improved the NAR retention in the stratum corneum (5-fold, *p* < 0.0001) compared to the control, and 3D device B enhanced drug penetration to the skin’s inner layers almost 3-fold (*p* < 0.0001). They are both mainly composed of PVA, corresponding to 70 and 44% of the device constitution. NAR accumulation can be facilitated by the hydrophobic nature of such a polymer, which may interact with the stratum corneum and decrease its overall lipophilicity. Such a hypothesis correlates with the obtained results, in which the device that contains the higher PVA content (3D device A), significantly enhances NAR accumulation in the skin’s outmost superficial layer (Figure 8A), almost preventing the drug from reaching the receptor compartment (*p* < 0.001, Figure 8C).

In contrast, while device B’s PVA content was insufficient to restrain NAR permeation through the skin compared to control, it was probably responsible for a delayed permeation, meaning that, with the same experimentation time, more drug accumulated in the viable skin layers. Hence, 3D device B did not control the drug release but most certainly interacted with the skin, resulting in a slower drug penetration profile, which could also signify an enhanced local effect. Further clinical studies could precisely estimate the loading drug doses ideal for attaining such an enhanced local effect.

Although FDM printing has become popular, the use of this technology for producing topical delivery systems has been limited to a few studies [30,31,32], with more examples of intradermal implants and scaffolds developed [33,34,35,36]. To the best of our knowledge, this is the first time that an FDM printed device intended for a local dermatological effect has been successfully produced and tested in a porcine ear skin model for permeation experiments.

## 4. Conclusions

Patient-centered drug devices are promising since personalized therapeutic strategies can address individual patients’ unique physiological needs, making treatment more effective. In this regard, using 3DP drug-delivery wearable devices as a topical treatment for aesthetic perforation complications is a feasible and innovative option. Notably, using predictive models based on statistical strategies such as the mixture design proved very useful in obtaining an adequate filament composition, i.e., with printability, stability, and the desired drug release profile, allowing the use of a broader range of 3DP materials. Finally, selected filaments demonstrated matrix physical-chemical stability, targeting the drug delivery to the superficial skin layers with promising prospects for topical treatment.

## Figures and Tables

**Figure 1 pharmaceutics-16-00077-f001:**
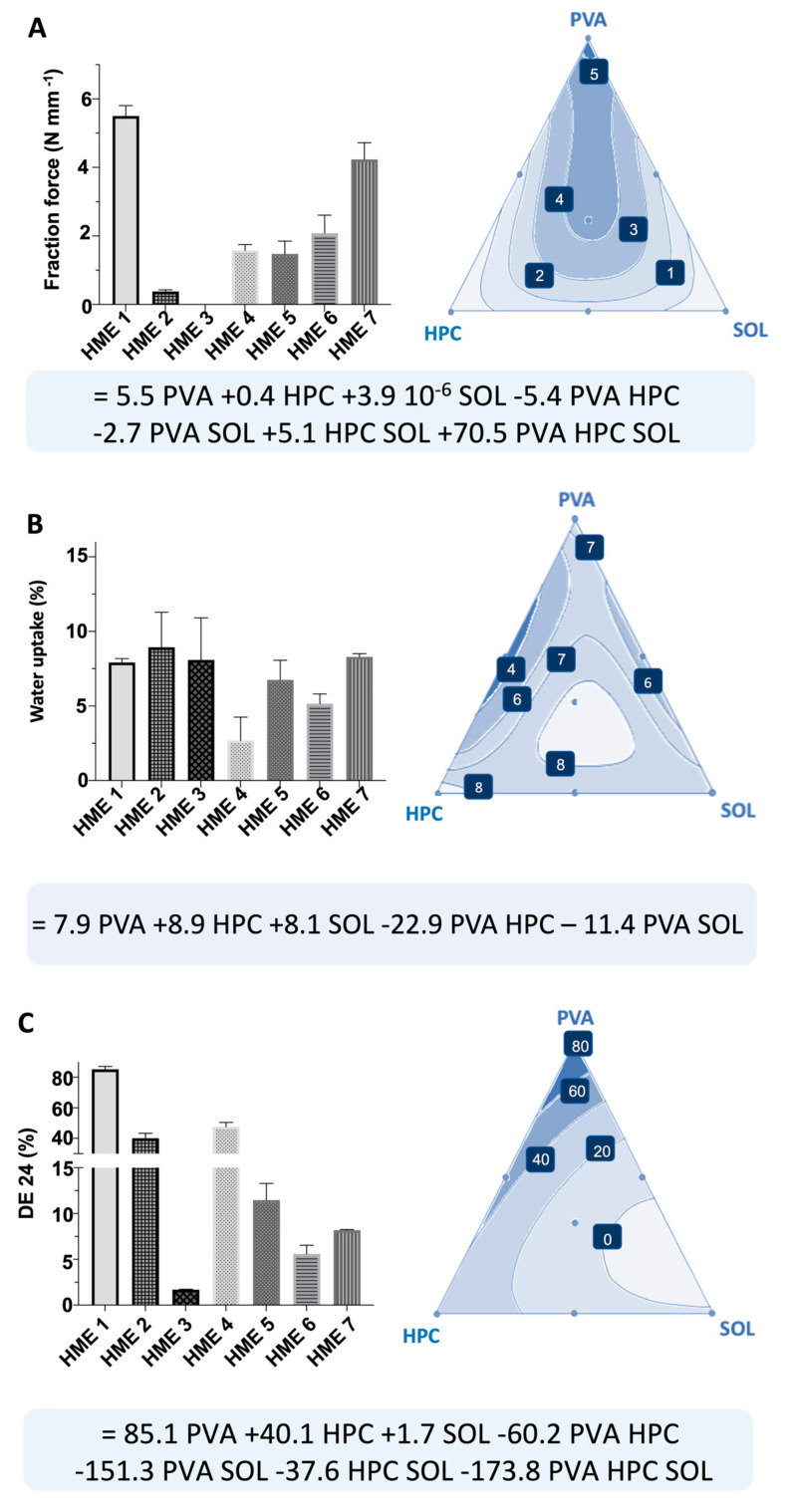
(**A**) Fraction force, (**B**) water uptake, and (**C**) dissolution efficiency (DE24) of filaments results, together with the response surface and predictive equation.

**Figure 2 pharmaceutics-16-00077-f002:**
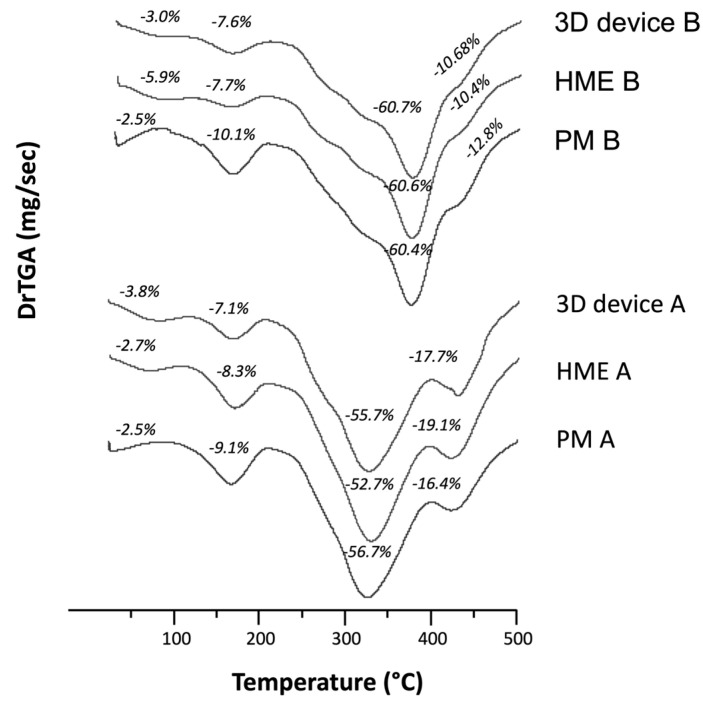
First-derivative TGA curves (DrTGA) of samples before treatment (PM), after hot-melt extrusion (HME), and after printing (3D device) for the formulations A and B. All weight-loss events described in the 25–500 °C range are indicated in the curves as a percentage (%).

**Figure 3 pharmaceutics-16-00077-f003:**
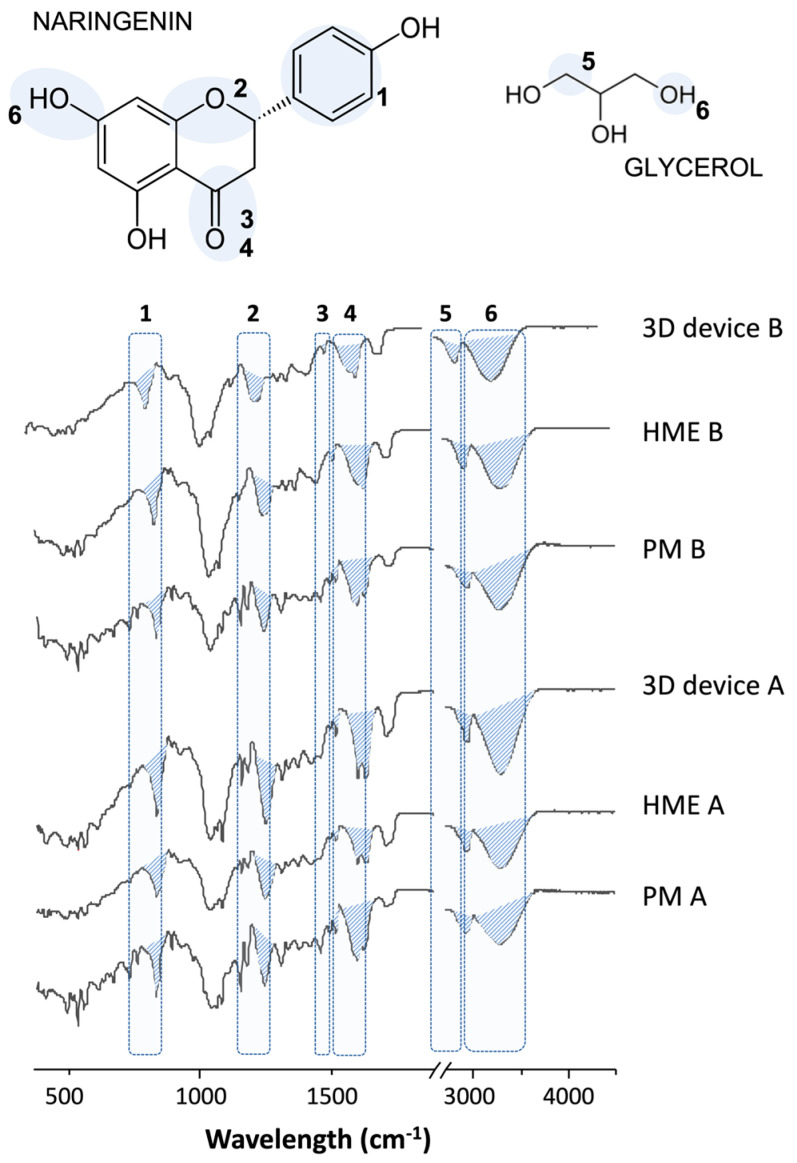
Fourier transform infrared (FTIR) spectra of the physical mixtures (PM), extrudates (HME), and printed samples (3D device) for the formulations A and B. Peaks related to the naringenin and glycerol functional groups are highlighted in blue in the spectra and correlated with their chemical structure.

**Figure 4 pharmaceutics-16-00077-f004:**
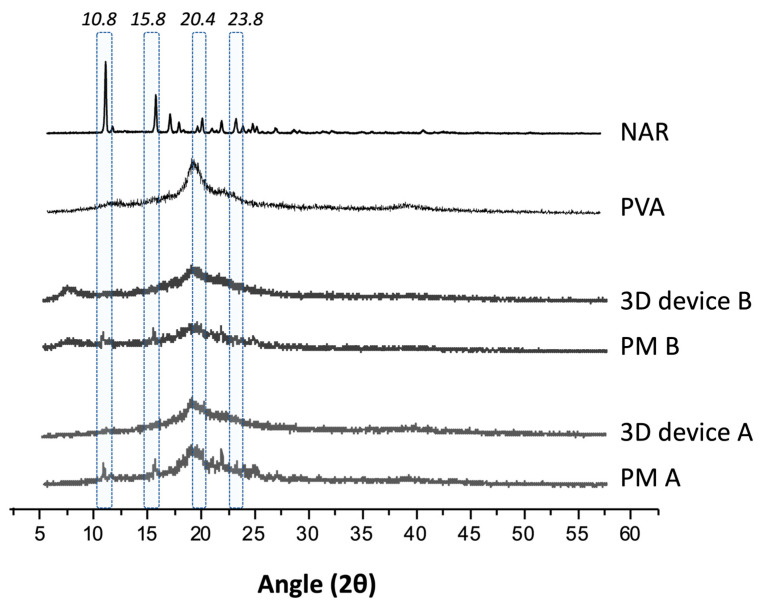
X-ray powder diffraction profile of naringenin (NAR) as supplied, polyvinyl alcohol (PVA), which is a semi-crystalline polymer, physical mixtures (PM), and printed devices (3D device) for the formulations A and B. NAR characteristic peaks are highlighted in blue.

**Figure 5 pharmaceutics-16-00077-f005:**
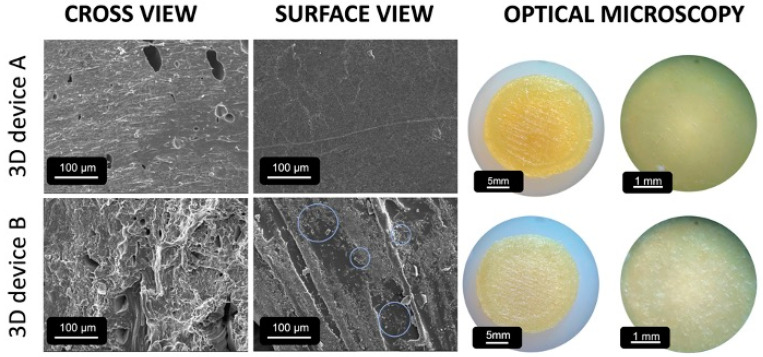
Scanning electron microscope images and optical microscopy photomicrographs of printed devices produced with filaments A and B. On the surface of filament B, crystal clusters are indicated by blue circles.

**Figure 6 pharmaceutics-16-00077-f006:**
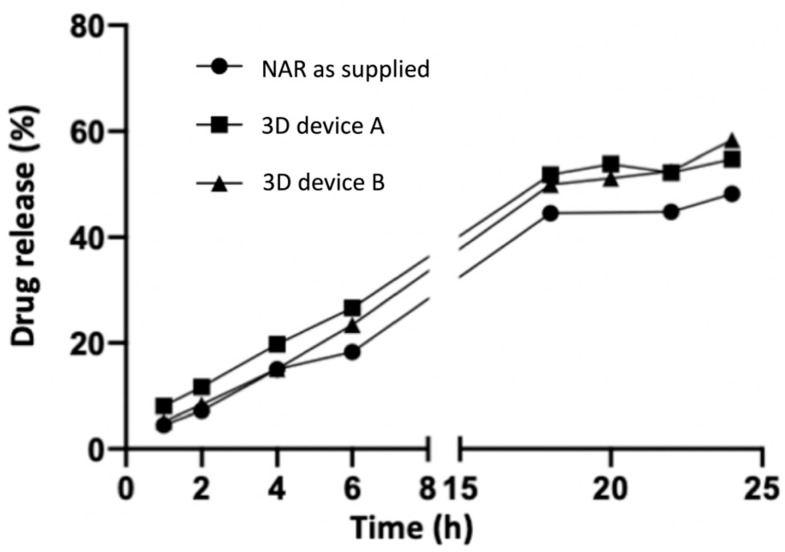
Drug release profile of disks containing naringenin dried solution (NAR) and printed disks (*n* = 5).

**Figure 7 pharmaceutics-16-00077-f007:**
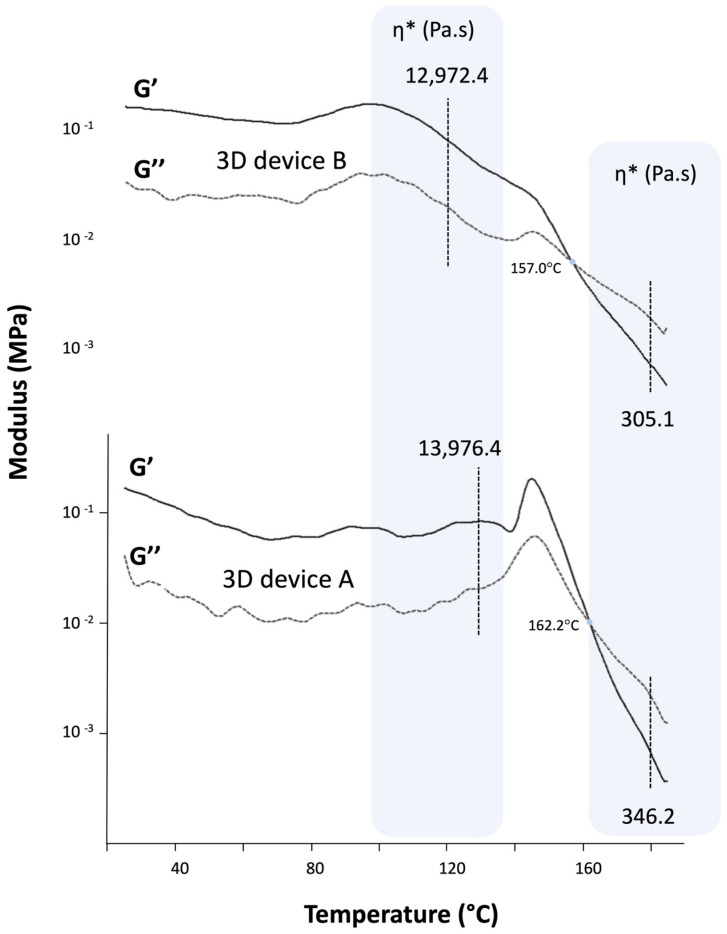
Dynamic oscillation temperature ramp of samples. The rheological patterns G′ and G″ are plotted, and the complex viscosity (η*) at HME processing temperature (130 °C for 3D device A and 120 °C for 3D device B) and printing temperature (180 °C) are indicated and shaded in blue. The crossover temperature (G′ = G″) was 162.2 °C for 3D device A and 157.0 °C for 3D device B.

**Figure 8 pharmaceutics-16-00077-f008:**
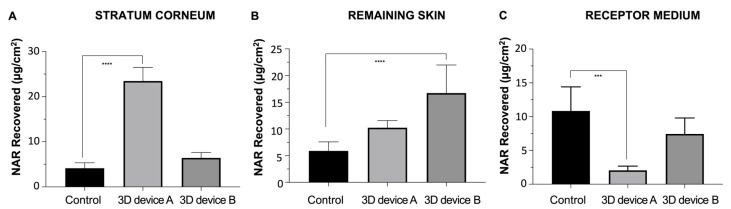
Naringenin (NAR) recovered from the (**A**) stratum corneum, (**B**) remaining skin, and (**C**) receptor medium (*n* = 5) (*** *p* ≤ 0.001; **** *p* ≤ 0.0001) from the drug solution control and the 3D devices A and B.

**Table 1 pharmaceutics-16-00077-t001:** System composition and extrusion conditions together with photomicrography of the filaments.

Code	Polymer (%)	Extrusion Conditions	OpticalMicroscopy
PVA	HPC	SOL	Temperature (°C)	Rotation (rpm)
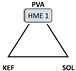	70.0	0.0	0.0	130	25	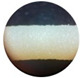
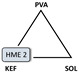	0.0	70.0	0.0	110	10	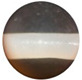
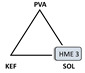	0.0	0.0	70	120	50	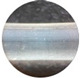
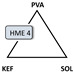	35.0	35.0	0.0	120	25	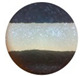
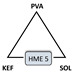	0.0	35.0	35.0	120	25	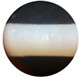
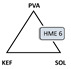	35.0	0.0	35.0	120	50	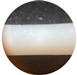
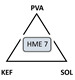	23.3	23.3	23.3	120	40	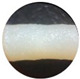

**Table 2 pharmaceutics-16-00077-t002:** Selected optimized conditions and the composition of the extrudates chosen together with the response desirability.

Identification	Water Uptake	Fraction Force	DE24	Composition (%, *w*/*w*)					Desirability
				NAR	GLY	SOL	HPC	PVA	
Filament A	Minimize	maximize	maximize	15	15	0	0	70	0.68
Filament B	Minimize	maximize	target 40	15	15	7	19	44	0.70

## Data Availability

The data presented in this study are available in this article.

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
