# Peer review of "Customizable Three-Dimensional Printed Earring Tap for Treating Affections Caused by Aesthetic Perforations"

_pharmaceutics, 2024, doi:10.3390/pharmaceutics16010077_

Round 1
Reviewer 1 Report
Comments and Suggestions for Authors
In this work the authors have developed 3D printed customised ear taps using a range of extruded polymers. The work is very interesting and shows the capabilities of FDM printing.
The authors have done a large number of experimental work and provide full characterisation of the prodded filaments including the ex-vivo behaviour of the printed structures. The discussion is in agreement with the experimental findings and the authors have provided sufficient justification.
I have minor query regarding the small crystals observed on the second filament. Did the authors investigate if the crystallisation continues under stability and the drug turns to crystalline from the amorphous state? This would affect the printability of the filaments but also the performance of the printed taps. It would be good if the authors add a couple of lines.
Author Response
In this work the authors have developed 3D printed customised ear taps using a range of extruded polymers. The work is very interesting and shows the capabilities of FDM printing.
The authors have done a large number of experimental work and provide full characterisation of the prodded filaments including the ex-vivo behaviour of the printed structures. The discussion is in agreement with the experimental findings and the authors have provided sufficient justification.
Response: Thank you for the positive evaluation of our manuscript.
I have minor query regarding the small crystals observed on the second filament. Did the authors investigate if the crystallisation continues under stability and the drug turns to crystalline from the amorphous state? This would affect the printability of the filaments but also the performance of the printed taps. It would be good if the authors add a couple of lines…
Response: The stability of pharmaceutical printed systems, particularly in the physical aspect mentioned by the reviewer, is highly relevant. Stability studies are necessary for the commercialization of these devices. However, this work is still a proof-of-concept study. We appreciate your suggestion, and accordingly, we commented on this issue in lines 375-378 of the revised manuscript.
“Moreover, physical modifications to the filament, such as the crystallization of components, can jeopardize its printability. Consequently, the rheological properties of the extruded formulations were measured and correlated with printability in this study.”
Reviewer 2 Report
Comments and Suggestions for Authors
This paper delineates the utilization of 3D printing to create drug-eluting devices aimed at treating prosthetic infections. Two devices, incorporating the anti-inflammatory model drug naringenin (NAR), were manufactured and assessed, showing both rapid and slow drug release properties. The manuscript is recommended for publication due to its comprehensive experimental validation and well-documented outcomes. Here are specific comments the author might consider for revising their manuscript:
1. Define each abbreviation upon its initial usage.
2. Determine the filament's phase transition and thermostability.
3. Did the authors thoroughly investigate and analyze the printability of the devices?
4. The release profiles of devices A and B appear similar. Can the authors elaborate on the reasons behind this similarity?
5. Is it possible that the authors meant "infection" instead of "affection"?
6. Enhance the introduction by providing more contextual information about prosthetic infections.
Author Response
This paper delineates the utilization of 3D printing to create drug-eluting devices aimed at treating prosthetic infections. Two devices, incorporating the anti-inflammatory model drug naringenin (NAR), were manufactured and assessed, showing both rapid and slow drug release properties. The manuscript is recommended for publication due to its comprehensive experimental validation and well-documented outcomes. Here are specific comments the author might consider for revising their manuscript:
Response: We appreciate the careful evaluation and comments on our manuscript. Please find below all your comments listed and addressed one-by-one. The changes in the manuscript are highlighted in yellow in the revised manuscript.
- Define each abbreviation upon its initial usage.
Response: Done. All occurrences of abbreviated terms were checked.
- Determine the filament's phase transition and thermostability.
Response: The composition of the filaments, especially the complex formulation of filament B, hampers the determination of the glass transition temperature (Tg) of the polymers by the techniques used in this work, i.e., thermal analysis and oscillatory rheology.
This occurs because filament A is composed of a semi-crystalline polymer (PVA), and therefore, the percentage of amorphous material is naturally reduced, making the identification of Tg challenging. In fact, the variation in heat capacity during the glass transition is so small that a large amount of material is required to measure the temperature range corresponding to this thermal event by DSC, even for pure polymer.
This problem is even more pronounced for formulation B, as three different polymers are used, with amorphous polymers (SOL and HPC) representing 26% of the formulation and the semi-crystalline polymer (PVA) 44%. Additionally, as observed by TGA analysis, the loss of water and evaporation of the plasticizer begin at approximately 40°C and 100°C, respectively. These endothermic events appear in the DSC curve, making it impossible to identify any change in the characteristic baseline of Tg. Moreover, eliminating the thermal history of the sample would alter the polymer plasticization, and the identified Tg would not be accurate.
Furthermore, due to the acquisition machinery's architecture (plate-plate geometry), the rheological analysis is more suitable for molten materials. This is why the analysis ramp was conducted decreasingly, heating the material from 185°C and then acquiring deformation data during cooling. Consequently, this analysis is ideal for evaluating filament deformation in the melt state, representing the material exposed to a new heating process (3D printing). Therefore, the performed analysis is unsuitable for identifying the glassy state, thus preventing the identification of polymer relaxation from the rubbery to the glassy state. In fact, we tried to determine Tg from both techniques; however, in this scenario, it was impractical to identify it for filaments A and B.
Regarding thermal stability, we have incorporated further discussion into the manuscript based on the TGA data. Please check the revised manuscript, lines 316-322
“No discernible differences in degradation curves were observed between the physical mixture, extruded filament, and 3D-printed devices. This observation suggests that the thermal and mechanical stresses from extrusion, along with the subsequent thermal stress during printing, did not induce degradation or any consequential interaction that might have adversely affected the components of the sample. Therefore, under the processing conditions used, the formulation demonstrated thermal stability without any signs of incompatibility or material loss.”
- Did the authors thoroughly investigate and analyze the printability of the devices?
Response: The printability of filaments was investigated through oscillatory rheology analyses, with the filament in its molten state, as, in the 3D printer, this material undergoes heating until it reaches the melting point, and its subsequent deformation plays a crucial role in the production of printed objects. To enhance understanding, we have expanded the discussion regarding rheological analysis to provide further clarity on this matter. Please check the revised manuscript, lines 369-373.
“Rheological analysis of the molten material is typically used to evaluate the printability of polymeric filaments in an FDM 3D printer [18]. Specifically, printability assessment involves monitoring the complex viscosity and storage and loss moduli during the printing temperature. The filaments should have a balance between flexibility and brittleness to feed an FDM 3D printer properly.”
- The release profiles of devices A and B appear similar. Can the authors elaborate on the reasons behind this similarity?
Response: The 3D devices were designed to slowly disintegrate to keep the earring tap in contact with the inflamed skin for a prolonged period. Although devices A and B have different solubilization capacities, in the drug release assay, the limited free water and the direct contact of the devices with the membrane delay their disintegration, leading to similar drug release kinetics. As suggested by the reviewer, the discussion about this was added to the revised manuscript (lines 362-365).
“Although 3D devices A and B have different solubilization capacities, in the drug release assay, the limited free water and the direct contact of the devices with the membrane delay their disintegration, leading to similar drug release kinetics.”
- Is it possible that the authors meant "infection" instead of "affection"?
Response: The term "affection" is broader than "infection", being synonymous with an abnormal condition of the body (Merriam-Webster dictionary). It seems more appropriate here as the potential treatment of the 3D earring could be aimed at infection, inflammation, or pain.
- Enhance the introduction by providing more contextual information about prosthetic infections.
Response: We thank the revisor for the suggestion. Accordingly, some information about prosthetic infections has been added to the introduction. Please check the revised manuscript in lines 43-44 and 57-63.
“… even devices that can treat prosthetic infections, a therapeutic intervention fraught with numerous challenges within contemporary medical and pharmaceutical domains [5,6,7].”
“Recognizing the limitations of conventional approaches, recent advancements in medical technology have introduced the potential of 3D printed devices to address complications arising from prosthetic infections [6,7]. In fact, navigating the landscape of pharmaceutical interventions for addressing prosthetic infections in the contemporary era brings forth an array of challenges and intricacies, necessitating innovative resolutions enabled by the advancements in 3D printing technology.”
Reviewer 3 Report
Comments and Suggestions for Authors
1. An upgrade in the resolution of the manuscript figures is necessary as the resolution is too low to discern details effectively.
2. The unit "Mpa" in the x-axis of Figure 7 needs to be corrected to "MPa."
3. What is the reason for not conducting cytotoxicity evaluation? In vitro tests using skin cells (fibroblasts and keratinocytes, etc.) should be performed.
4. Since in vivo tests were not conducted in this study, the conclusion needs to be revised.
- "Finally, the characterization of … enabling topical treatment."
5. A comprehensive English correction for the manuscript is required.
Comments on the Quality of English LanguageA comprehensive English correction for the manuscript is required.
Author Response
- An upgrade in the resolution of the manuscript figures is necessary as the resolution is too low to discern details effectively.
Response: The figures were sent separately in the submission system in high quality. Moreover, higher-quality figures were incorporated into the revised manuscript as required. Please see the revised manuscript.
- The unit "Mpa" in the x-axis of Figure 7 needs to be corrected to "MPa."
Response: Done. Thank you for the correction. Please see the revised figure 7.
- What is the reason for not conducting cytotoxicity evaluation? In vitro tests using skin cells (fibroblasts and keratinocytes, etc.) should be performed.
Response: Thanks for the valuable suggestion.
Cell culture assays using fibroblasts and keratinocytes are very interesting for evaluating the safety of a topical device such as the one proposed in the article. However, the study presented here is still proof of concept of an innovative system. Thus, the authors prioritized evaluating physicochemical characteristics and investigating the drug distribution profile in the skin. Despite not conducting cytotoxicity studies, the polymeric matrices chosen are of pharmaceutical grade and, therefore, biocompatible. This information was included in the revised manuscript (lines 94-95).
“The HME filaments were prepared according to a simplex centroid mixture design with three components without constraints to determine the ideal combination of biocompatible polymers”
- Since in vivo tests were not conducted in this study, the conclusion needs to be revised.
- "Finally, the characterization of … enabling topical treatment."
Response: Thank you for the correction. Please see the revised manuscript (line 458).
“Finally, selected filaments demonstrated matrix physical-chemical stability, targeting the drug delivery to the superficial skin layers with promising prospects for topical treatment.”
- A comprehensive English correction for the manuscript is required.
Response: An English review was carried out throughout the manuscript.
Round 2
Reviewer 2 Report
Comments and Suggestions for Authors
The authors have revised the manuscript